# The Zagreb (Croatia) M5.5 Earthquake on 22 March 2020

**Snježana Markušić [1,\*], Davor Stanko [2], Tvrtko Korbar [3], Nikola Belić [3], Davorin Penava [4] and Branko Kordić [3]**

[1] Department of Geophysics, Faculty of Science, University of Zagreb, 10000 Zagreb, Croatia

[2] Faculty of Geotechnical Engineering, University of Zagreb, 42000 Varaždin, Croatia; davor.stanko@gfv.unizg.hr

[3] Department of Geology, Croatian Geological Survey, 10000 Zagreb, Croatia; tkorbar@hgi-cgs.hr (T.K.); nbelic@hgi-cgs.hr (N.B.); bkordic@hgi-cgs.hr (B.K.)

[4] Faculty of Civil Engineering and Architecture Osijek, Josip Juraj Strossmayer University of Osijek, 31000 Osijek, Croatia; dpenava@gfos.hr

\* Correspondence: markusic@gfz.hr; Tel.: +385-1-460-5913

**Abstract:** On 22 March 2020, Zagreb was struck by an M5.5 earthquake that had been expected for more than 100 years and revealed all the failures in the construction of residential buildings in the Croatian capital, especially those built in the first half of the 20th century. Because of that, extensive seismological, geological, geodetic and structural engineering surveys were conducted immediately after the main shock. This study provides descriptions of damage, specifying the building performances and their correlation with the local soil characteristics, i.e., seismic motion amplification. Co-seismic vertical ground displacement was estimated, and the most affected area is identified according to Sentinel-1 interferometric wide-swath data. Finally, preliminary 3D structural modeling of the earthquake sequence was performed, and two major faults were modeled using inverse distance weight (IDW) interpolation of the grouped hypocenters. The first-order assessment of seismic amplification (due to site conditions) in the Zagreb area for the M5.5 earthquake shows that ground motions of approximately 0.16–0.19 g were amplified at least twice. The observed co-seismic deformation (based on Sentinel-1A IW SLC images) implies an approximately 3 cm uplift of the epicentral area that covers approximately 20 km². Based on the preliminary spatial and temporal analyses of the Zagreb 2020 earthquake sequence, the main shock and the first aftershocks evidently occurred in the subsurface of the Medvednica Mountains along a deep-seated southeast-dipping thrust fault, recognized as the primary (master) fault. The co-seismic rupture propagated along the thrust towards northwest during the first half-hour of the earthquake sequence, which can be clearly seen from the time-lapse visualization. The preliminary results strongly support one of the debated models of the active tectonic setting of the Medvednica Mountains and will contribute to a better assessment of the seismic hazard for the wider Zagreb area.

**Keywords:** Zagreb earthquake; Medvednica Mountains; seismicity; amplification; structural modeling; building damage

## 1. Introduction

The city of Zagreb is the Croatian capital and is situated in the contact area of three major regional tectonic units: the Alps in the northwest, the Pannonian Basin in the east and the Dinarides in the south. The causes of earthquakes are tectonic movements that occur in the upper crust because

of interactions between the underlying lithospheric plates: the European plate and the Adriatic microplate (e.g., see [1–3]). As a result of the compression and/or subduction of the plates, the upper crustal faults become seismic sources of earthquakes.

Earthquakes and seismic activity in the wider Zagreb area are not uncommon. At the end of the 19th century, Josip Mokrović [4], a well-known Croatian geophysicist, calculated that Zagreb had been shaken by earthquakes as many as 661 times from 1502 to 1883. The strongest earthquake in recent Zagreb history occurred in 1880 and has been estimated according to macroseismic observations at magnitude 6.3 [5]. However, the magnitude 5.5 earthquake that occurred on Sunday morning 22 March 2020 was the strongest instrumentally recorded seismic event in Zagreb since Andrija Mohorovičić established the first seismograph in 1908.

The city is situated in the southern foothills of the Medvednica Mountains which are uplifted along the SW–NE-striking Žumberak–Medvednica–Kalnik fault zone. The earthquakes in the area are the result of the interface between crustal fragments bordered by active faults [6,7]. Matoš et al. [8] concluded that the most tectonically active areas are located at the SW corner and in the central part of the Medvednica Mountains, where they are likely related to the longitudinal steeply southeast-dipping reverse faults and transversal strike-slip/normal faults, respectively.

The earthquake mechanisms reveal predominantly N–S directed P-axes in the study area that indicate the prevalence of compressional tectonics with reverse faulting [7]. These data are in agreement with stress calculations and kinematics of Quaternary structures obtained from geological studies [6,9]. However, while Prelogović et al. [6] suggested an active longitudinal transpressive positive flower structure of the Medvednica Mountains bordered by steep divergent reverse faults, Matoš et al. [8] indicated unidirectional top-to-the-north steep reverse faults along the Medvednica Mountains as the main earthquake sources.

## 2. Geological and Tectonic Overview

The Medvednica Mountains, a NE–SW-striking mountain range within the Internal Dinarides, is a marginal Pannonian inselberg located in a geotectonically interesting area where kinematic interactions of neighboring orogens are observed. The Alps and Dinarides are orogens with different subduction polarities: Adria the upper plate in the Alps versus Adria the lower plate in the Dinarides (e.g., see [10]). A complex active tectonic regime in the transitional area between the Southeastern Alps, Northwestern Dinarides and Tisza Mega-unit of the Pannonian Basin is a result of the interaction of the upper crustal tectonic blocks formed during the Mesozoic to Cenozoic evolution of the area [11]. Its present-day position and trend are explained by an eastward displacement (tectonic escape) and an approximately 130° clockwise rotation of the tectonic block comprising the Medvednica Mountains and its surrounding inselbergs during the period from the latest Paleogene to the earliest Neogene [9,12].

The Medvednica Mountains are composed of a Paleozoic to Mesozoic metamorphic core overlain by non-metamorphic Mesozoic and Cenozoic rocks ([13–15]; Figure 1a). All units are truncated by normal and reverse faults and are surrounded by the Miocene–Quaternary deposits of the Pannonian Basin.

Van Gelder et al. [11] demonstrated that the area of the Medvednica Mountains, situated near neighboring orogens with different subduction polarities, i.e., between the Alps and the Dinarides, has a complex and dual tectonic history. The zone of interference is large; not only do the structures of the Dinarides extend far into the Southern Alps, but the structures of the Eastern and Southern Alps also extend much farther into the Dinarides than previously thought. To show the structural complexity of the Medvednica Mountains, Van Gelder et al. [11] interpreted two cross-sections across the mountain (Figure 1b). The latest tectonic phase of the tectonic evolution of the Medvednica Mountains is characterized by a system of Pliocene to Quaternary southeast-dipping reverse faults (D6 faults in Figure 1b). The Northern Medvednica Boundary Fault [16] is the major fault of the system and is probably still active [8].

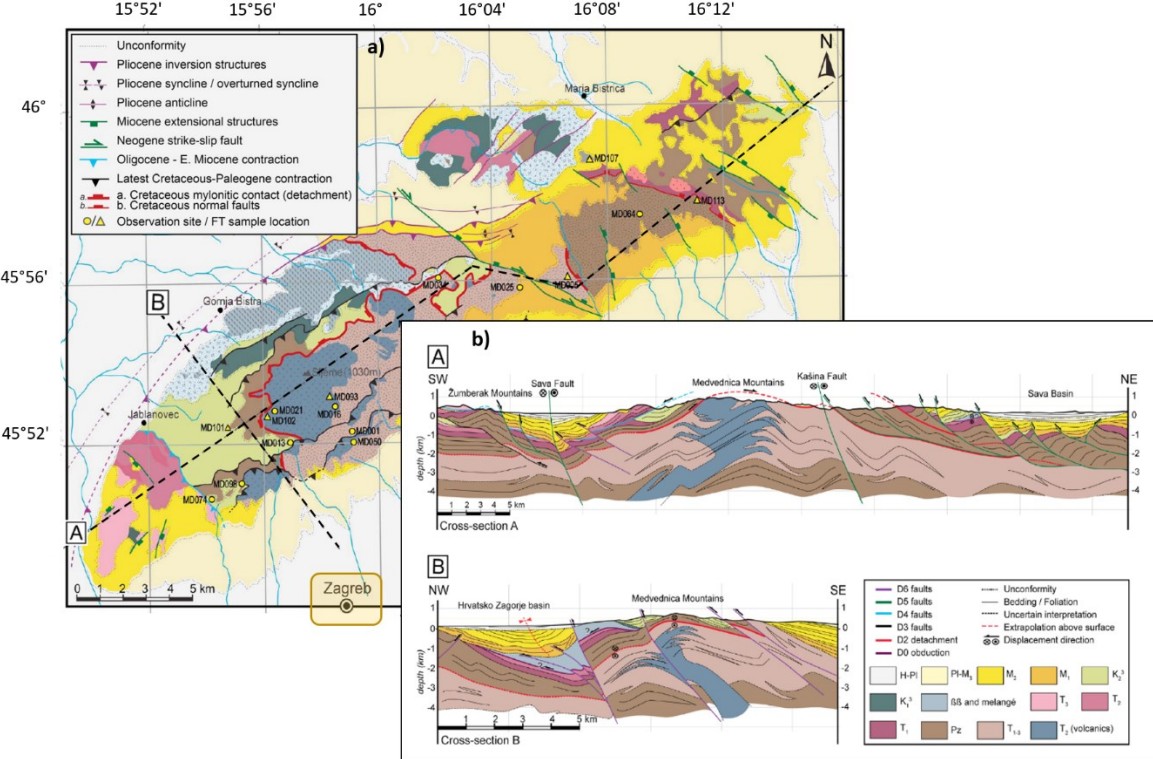

**Figure 1.** (**a**) Geological map of the Medvednica Mountains. The map shows the location of the cross-sections A and B; (**b**) Cross-sections across the Medvednica Mountains: (A) longitudinal (SW–NE) and (B) transversal (NW–SE) (after [11]).

## 3. Seismicity of the Wider Zagreb Area

Systematic data collection for earthquakes in Croatia, and consequently in Zagreb, started in the 19th century with the magazine Luna (which was published in German) where earthquakes that occurred in Zagreb after 1872 were recorded. There were many earthquakes in and around Zagreb, but for those that occurred before 1880, descriptions are scarce, so the actual magnitudes of the earthquakes cannot be estimated and the locations of the epicenters are unreliable. The first comprehensive consideration of Zagreb earthquakes is in the annual reports published by M. Kišpatić in 1879 [17], which include data on all earthquakes that occurred in Zagreb from 1502 to 1879. The strongest earthquakes with epicenters around Zagreb are mentioned here, based on the "Croatian Earthquake Catalogue" (CEC; updated version first described in [18]).

The earthquake that struck on a chilly Tuesday on 9 November 1880 at 07:03 UTC, with a magnitude of 6.3, is the strongest earthquake that occurred in the Zagreb epicentral area. Its hypocenter was in the Medvednica Mountains, close to locations of Kašina and Planina. The newspapers wrote that Zagreb had shaken, frightening people and causing two casualties [5]. According to historical records, at this time, Zagreb had slightly fewer than 30,000 inhabitants and approximately 2500 buildings. A total of 1400 buildings were damaged or destroyed by the earthquake.

It is understandable that such a devastating earthquake caused panic among the population, especially since numerous weaker earthquakes followed, especially in the next six months. There were quite a number of people who spread alarming news, such as the idea that Zagreb was on the verge of collapse because it lies above an underground volcano; they fabricated mud volcanoes, and some observed bluish flames at the top of the Medvednica Mountains and new hot springs in Stubica.

This earthquake was not only significant in terms of its catastrophic effects but also because it gave impetus to the study of earthquakes and marks the time when this study actually began systematically. The belief was then founded that the study of earthquakes could not be the sole interest of individuals but that it was an important task needed for the whole community.

A pioneering role in this research was played by the Academy of Science and Art and the Meteorological Observatory in Zagreb, from which, thanks to Andrija Mohorovičić, the world-renowned "Zagreb Seismological School" developed.

After the 1880 earthquake, on 17 December 1901, a strong earthquake (with a magnitude of 4.6) occurred with an epicenter in the vicinity of Šestine. In almost the same location as the 1880 earthquake, two more strong earthquakes occurred on 17 December 1905 and 2 January 1906.

Regarding the recent seismic activity in the wider Zagreb area, the strongest earthquake occurred on 3 September 1990, with the epicenter near Kraljev Vrh and a magnitude of 5.0.

Figure 2 displays the locations of the epicenters of all earthquakes that have occurred in the wider Zagreb area (BC to 2019).

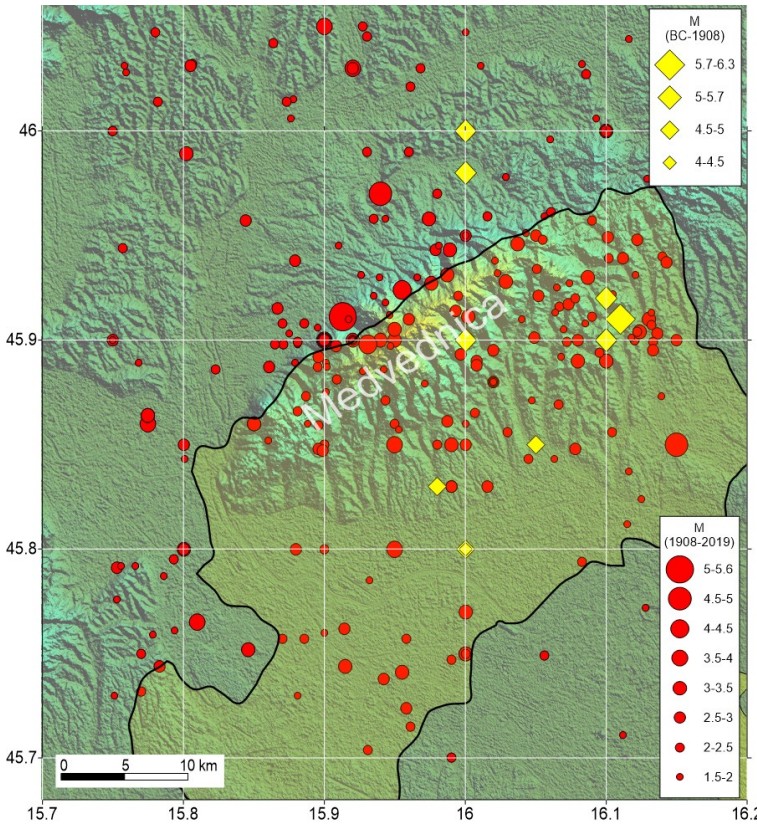

**Figure 2.** Spatial distribution of earthquakes with locations in the wider Zagreb area from BC to 2019 (according to the updated version of the Croatian Earthquake Catalogue, first described in [18]). The urbanized Zagreb city area is displayed with yellow shadowing.

## 4. The 22 March 2020 Earthquake

In the early Sunday morning of 22 March 2020, at 05:24 UTC, Zagreb residents were awakened by a strong earthquake. Its epicenter was 7 km north of the center of Zagreb, in the vicinity of Markuševac and Čučerje, and the hypocenter was at a depth of 10 km. In addition, it will be remembered as the strongest earthquake that has happened in the 140 years after the "Great Zagreb earthquake" of 1880. The earthquake was felt with a maximal intensity of VII–VIII °MSK (Figure 3) and had a magnitude ($M_L$) of 5.5. It was felt all over Croatia, even at a distance of more than 1000 km from the epicenter.

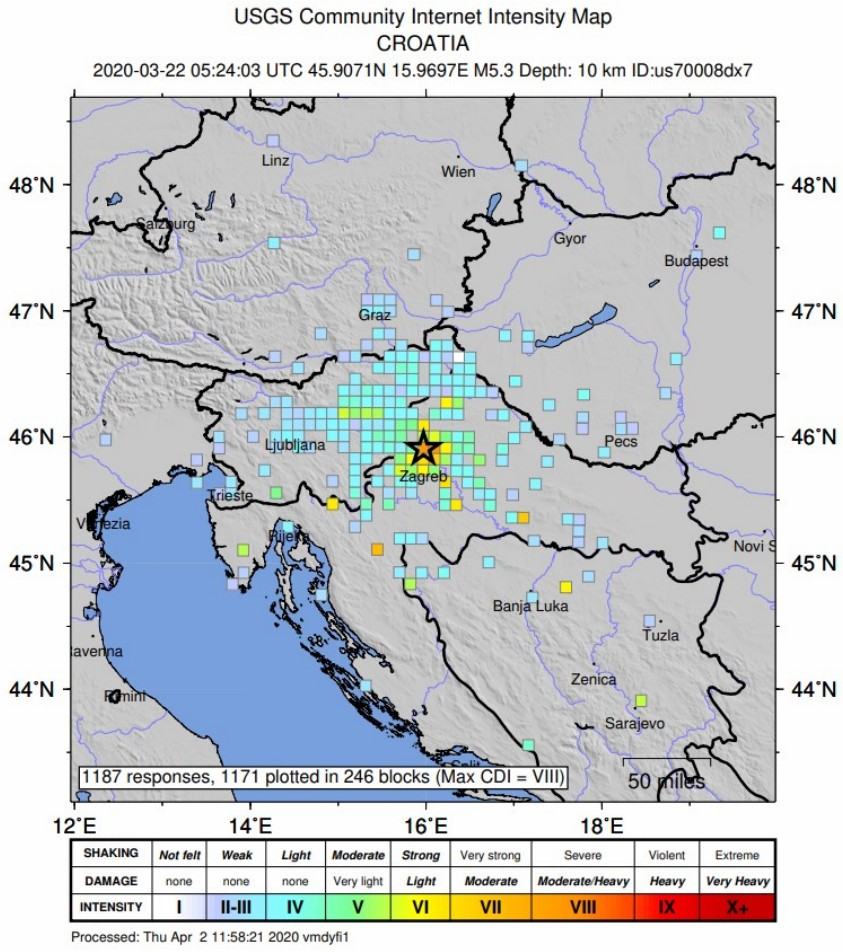

**Figure 3.** Intensity map according to data collected by USGS [19].

The second strong earthquake ($M_L$ = 4.9), occurred at 06:01 UTC. The third strong aftershock ($M_L$ = 3.7) was recorded at 06:41 UTC. In just over 24 hours since the main shock, 57 earthquakes with magnitudes greater than or equal to 2.0 occurred in Zagreb.

The M5.5 earthquake that struck Zagreb on 22 March 2020 caused more damage than would be expected for an earthquake of this magnitude. Therefore, shortly after the main shock, extensive multidisciplinary research was conducted.

Based on the analysis of data recorded in Croatia and around the world (waveform data taken from the USGS National Earthquake Information Center [20]), the focal mechanisms of the main earthquake and the strongest aftershock were calculated (Figure 4).

The fault-plane solution of the main shock indicates that the earthquake occurred on a reverse fault whose fault plane dips at an angle of 43° to the south-southeast or at an angle of 48° to the north-northwest. The axis of maximum tectonic pressure (P) was horizontal and in the SSE-NNW direction, while the axis of maximum tension (T) was almost vertical. The calculated fault mechanisms will be described in greater detail in Section 4.5 and linked to the geological features of the study area.

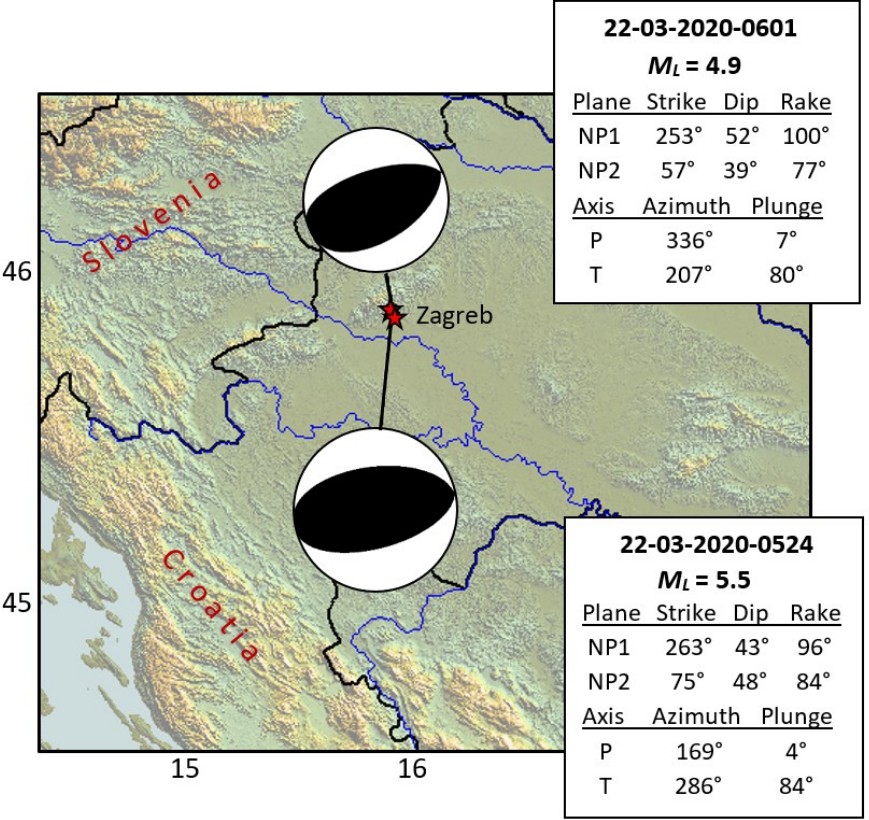

**Figure 4.** Fault-plane solutions of 22 March 2020 earthquakes at 05:24 UTC and 06:01 UTC. The focal sphere is shown in stereographic projection.

Since no one knows how much accumulated energy is released through a main earthquake, the aftershock activity is normal and not uncommon. Figure 5 presents the number of earthquakes of a certain magnitude (greater than 1.3) that occurred after the main earthquake of 22 March 2020 at 05:24 UTC.

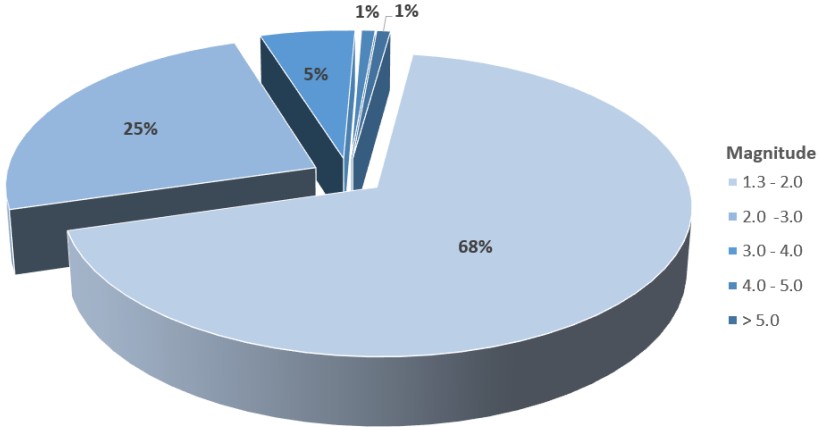

**Figure 5.** Percentage of earthquakes of each magnitude interval that occurred in the epicentral area of the Zagreb earthquake in the period from 22 March to 1 April 2020.

From the analysis of aftershocks in both the temporal and spatial domains and correlation with known geological features, it is possible to study the dynamics of the area under consideration in further detail. This is precisely what is presented in Section 4.5.

### 4.1. Damage Description

The earthquake hit the city center hardest, and the initial information indicated that the damage was great. Much damage to the cultural and historical architectural heritage of the city of Zagreb was reported, with severe damage to the south tower of Zagreb Cathedral. About 16,555 residential, public and monumental buildings were damaged (Figure 6) as declared by [21,22].

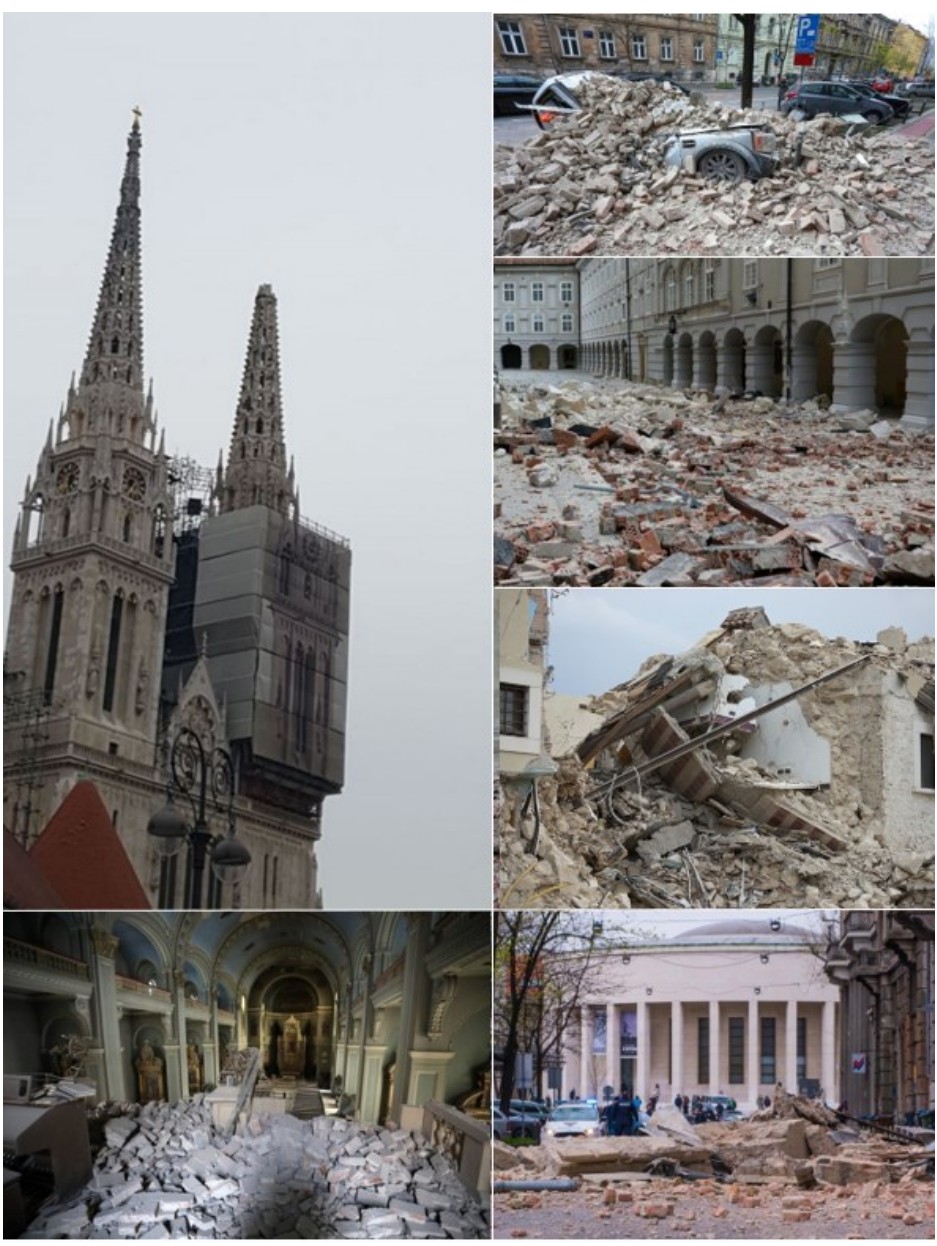

**Figure 6.** Photos of damage in Zagreb caused by the 22 March 2020 earthquake.

State institution buildings, sacred objects, museums, galleries, the Mirogoj Cemetery, the Rectorate, faculties, schools and theatres also suffered a great amount of damage.

The extent of the damage is also visible on the damage proxy map (DPM) (Figure 7), which was created by the Advanced Rapid Imaging and Analysis (ARIA) team at NASA's Jet Propulsion Laboratory and the California Institute of Technology, both in Pasadena, California. The map depicts areas that were likely damaged. The map was derived from synthetic aperture radar (SAR) images

from the Copernicus Sentinel-1 satellites, operated by the European Space Agency (ESA). The team compared a post-event image acquired on 23 March 2020 with pre-event images taken since January 2020. The image covers an area of 166 by 56 km (103 by 35 miles), shown by the large red polygon. Each pixel measures approximately 30 m across. The color variations from yellow to red indicate increasingly significant surface changes. Preliminary validation was performed by using photographs from various sources, including news media. This damage proxy map should be used as a guide to identify damaged areas. The full map is available online at [23].

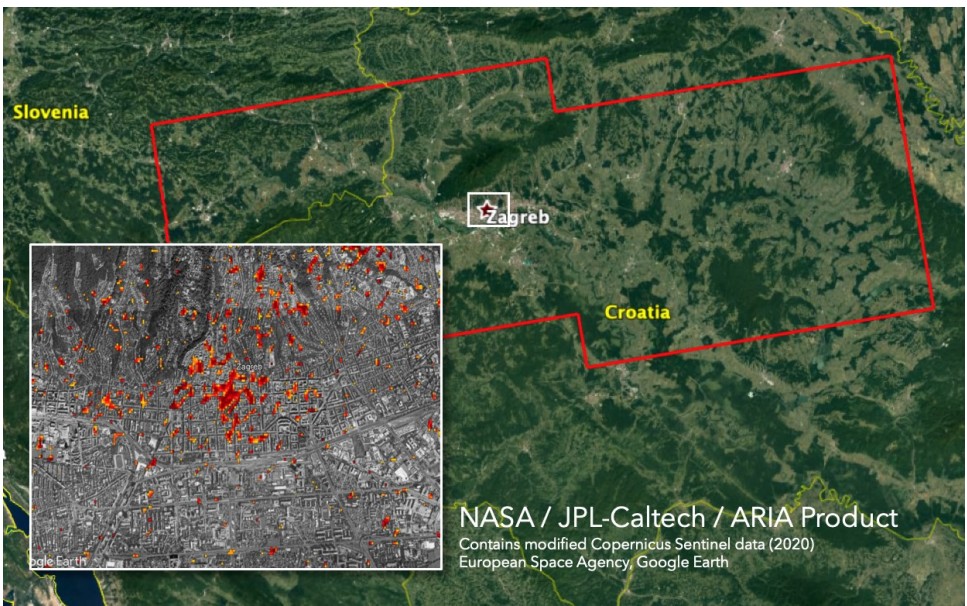

**Figure 7.** Damage proxy map created by the Advanced Rapid Imaging and Analysis (ARIA) team at NASA's Jet Propulsion Laboratory (JPL) and the California Institute of Technology. This map contains modified Copernicus Sentinel data, processed by the ESA and analyzed by the NASA JPL/Caltech ARIA team. This task was carried out at the JPL, funded by NASA. [23].

### 4.2. Building Performance

Figure 6 illustrates that in the old city center, three- to five-story masonry buildings, constructed from the time period of the Austro-Hungarian monarchy to 1920, sustained very heavy damage or were destroyed due to the M5.5 Zagreb earthquake on 22 March 2020. There were 26,197 damaged buildings in total (approximately 1900 uninhabitable), among which 9642 were family houses (as declared by [21,22]). As stated in the Disaster Risk Assessment for the Republic of Croatia [24], before 1920, masonry buildings used wooden slab construction, while reinforced concrete (RC) slabs were introduced in the period from 1920 to 1945. Before 1945, buildings possessed only initial earthquake resistance, and earthquake load was not considered in design (up to 5% of relevant peak ground acceleration - PGA was considered in design due to wind load). In compliance with Earthquake Hazard Map of Republic of Croatia [25], the relevant PGA based on 10% exceedance in 50 years, i.e., a return period of 475 years in the Zagreb area goes up to 0.28 $g$. The consideration of earthquake action in the design of buildings dates from 1946; however, the designs provided inadequate solutions. From 1945 to 1964, earthquake load was considered in the design of buildings by using simplified methodologies (the horizontal force at the top of the building); however, only up to 10% of the relevant PGA was considered in design. In 1964, as a result of the M6.9 Skopje (North Macedonia) earthquake in 1963, the Ordinance on Temporary Technical Regulations for Construction in Seismic Areas (Federal Republic of Yugoslavia (FRY) Official Journal, [26]) was issued. The ordinance required the construction of vertical or horizontal confining elements in masonry buildings, but it provided low earthquake resistance in general. In 1981, it was replaced by the Ordinance on Technical Standards for the Construction of High-Rise Buildings in Seismic Areas (FRY Official Journal, [27]), which provided moderate resistance. Between the years 1964 and 1998, 30–50%

of the relevant PGA was considered in the building earthquake resistance design, and 70–100% was considered between 1998 and 2013. The aforementioned regulations were completely replaced by the contemporary standard EN 1998-1 [28] in 2013, which ensured high earthquake resistance (i.e., the earthquake load considered equals 100% of the relevant PGA).

Buildings constructed after 1963 were found to have significantly better performance during the earthquake than buildings constructed before 1920 (or even 1945) [21,22,24], as assessed by structural engineers in the field [21,22,24]. The damage levels were sharply lower and thus related to different Modified Mercalli scale intensities (MM or MMI) and damage grades (DG), in compliance with [29]. Intensity scale provided the overall insight into the earthquake severity, but the performance of buildings in the field and their vulnerability was better judged based on the damage grade classifications, i.e., where Grade 1 (DG 1) indicates negligible to slight damage, Grade 2 (DG 2) indicates moderate damage, Grade 3 (DG 3) indicates substantial to heavy damage, Grade 4 (DG 4) indicates very heavy damage and Grade 5 (DG 5) indicates destruction. Damage grades could be regarded as a function of the earthquake intensity and provide a detailed description of damage resulting from an earthquake by highlighting specific building structural systems or elements (structural and nonstructural); therefore, they are the most adequate tool for the inspection and assessment of buildings damaged in earthquakes.

The aforementioned improvement was associated with the increased ductility demand of the regulations in the post-1963 period and the materials used for construction. Therefore, the time period of construction with respect to damaged buildings revealed dramatic differences in their performances. The unreinforced masonry (URM) buildings from the time period before 1920 (or even 1945) had damage several times greater than buildings with reinforced concrete (walls or infilled frames) or confined masonry from the post-1963 time period (very heavy damage and destruction, i.e., DG 4 and 5, versus negligible to slight and moderate damage, i.e., DG 1 and 2). The damage was accompanied by widespread collapse of brittle chimneys (they all had similar construction). The occurrence of chimney collapse is an indicator of intensity 8, or a severe earthquake, according to the Modified Mercalli scale (MMI 8). However, as stated in [30], dominant chimney collapse can be (and apparently often is) used as an indicator of intensity, so the damage should be assessed with all damage included and with chimney collapse excluded. In considering the latter, the earthquake damage in the large residential area of the Zagreb center can be classified as intensity 6, or a strong earthquake, according to the Modified Mercalli scale (MMI 6). By considering the time period and the construction material, the same principles as observed in the vicinity of the epicenter are consistent with the observations in the case of one- or two-story family houses. The key fact, as evident from available data, is that the measures of earthquake-resistant design substantially reduced the damage to buildings.

On the basis of the observed building damage, a subsequent retrofitting of the most sensitive and dangerous buildings (e.g., city center URM family houses, URM city center residential three- to five-story buildings and cultural heritage buildings) to reduce their vulnerability in compliance with the EN 1998-3 [31] provisions is required, particularly when an earthquake with M6.5 (MMI 8–9) is likely to occur (e.g., once every 475 years, [32]). Considering the structural weaknesses of the damaged buildings, especially those in pre-1964 structures, attention should be given to local and global strengthening and stiffening and unnecessary mass removal. In general, based on recommendations from [30], pre-1964 URM buildings can be retrofitted by, e.g., spraying concrete on the walls from inside to protect the outward appearance. However, heritage buildings require special attention due to their cultural significance and heritage conservancy requirements. An additional way to improve the performance of URM family houses is to insert confining elements into their structural wall intersections. The costs of building repair, estimated to be 6 billion US dollars, should also cover building performance improvement, at least for the most sensitive and dangerous buildings. Considering the case of the recent and nearby M6.4 Durres, Albania, 26 November 2019 earthquake [33] as an example, buildings with moderate and high levels of earthquake-resistant design experienced negligible to slight and moderate damage to RC components (DG 1 and 2). However, they also experienced substantial to heavy damage, including very heavy damage and

destruction of URM components (DG 3, 4 and 5). The damage resulting from the Durres, Albania, earthquake could be easily related to laboratory experiments, as described in [34,35]. If such an event affects Zagreb, it could have high social and economic consequences (human casualties, i.e., deaths and injuries, and financial costs) due to buildings collapsing or becoming mostly unserviceable.

*4.3. Assessment of Seismic Motion Amplification*

The intensity of earthquake shaking at certain sites in terms of observed or recorded strong ground motion is influenced by a complex system that depends on the source characteristics, attenuation of seismic waves and modification by the local site conditions (e.g., see [36,37]). The most important factors influencing the level of earthquake ground motion are the magnitude of the earthquake and the distance to its epicenter. However, local site effects, as the third important factor, can significantly amplify or de-amplify the level of shaking. The effects of local soil conditions or so-called "site effects" are defined as the modification of the incoming wavefield characteristics (amplitude, frequency content and duration) due to the specific geological site characteristics, geometrical features of the soil deposits and the surface topography. The modification is manifested as the amplification or de-amplification of ground motion amplitudes at all frequencies or periods at the surface, compared to the bedrock level [36]. Strong local site effects and different damage distributions for various local site conditions were observed in different areas during seismic events [38–42]. The ground response of the local site, i.e., the site amplification, can be classified in terms of "linearity", where amplification is proportional to the input ground motion, or in terms of "nonlinearity", where the soft soil responds as a strong damper of earthquake energy for strong input ground motion [43].

Interestingly, the first scientifically explained observation of the local soil amplification effects and variable earthquake damage due to different local geological units during shaking was presented by Stur [44], who analyzed the Klana (Croatia) earthquake sequence with the main shock on 1 March 1870 ($I_{max}$ = VIII °MSK), presenting a detailed report on building construction and observations of the unequal distribution of earthquake damage related to soil conditions with geological sketches, damage locations of objects and a map of the shaken area. Herak et al. [45] presented their macroseismic study of this earthquake sequence and measured the horizontal-to-vertical spectral ratios (HVSRs) of ambient noise at six locations to compare estimates of soil response in the epicentral area with observations of site effects [44] during the Klana earthquake. Presented findings and observations from [45] are similar to the observations of [44] about local site effects in Klana and Studena. Similarly, we attempted to assess the seismic site amplification distribution that occurred during the Zagreb M5.5 earthquake to possibly correlate it with observed damage in the city area and local site effects. For the first-order assessment of seismic site conditions, we used maps of the average shear velocity down to 30 m ($V_{S30}$), which is correlated with topographic slope [46], as displayed in Figure 8.

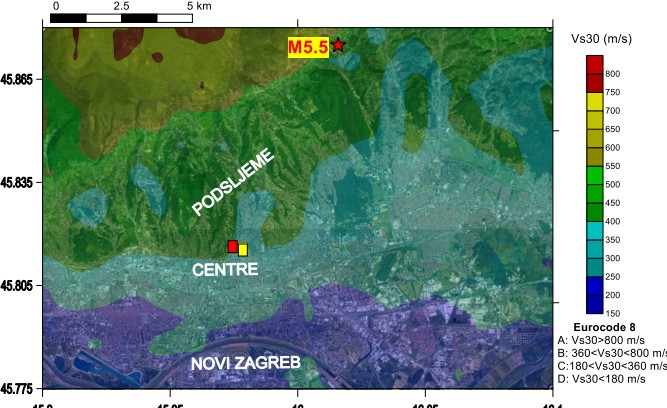

**Figure 8.** $V_{S30}$ map of the Zagreb area obtained based on approximation via correlation to topographic slope ([46]; source: [47]). Locations of Zagreb Cathedral (yellow square) and Croatian Parliament (red square) are shown. The M5.5 epicenter location is marked with a red star.

We used a nonlinear site amplification model developed by [48] based on the site parameter ($V_{S30}$) and the intensity of input rock motion ($PGA_{ROCK}$), similar to models developed by [49–51]. Based on the well-known properties of random vibration theory (RVT) and seismological source theory [52], $PGA_{ROCK}$ values for the M5.5 Zagreb earthquake were estimated (from the Fourier amplitude spectrum (FAS)). The input motion in the RVT-based method is typically defined by Brune's [53] omega-squared point-source stochastic model. The stress drop was set to the prescribed constant value of 100 bar (e.g., see [54]), the focal depth was taken as $h$ = 10 km, the average value of crustal shear wave velocity was equal to 3.5 km/s and the density was 2800 kg/m$^3$. Frequency-dependent attenuation, the $Q(f)$ value, was adopted from [55] as $Q_c(f) = 78f_c^{0.69}$, and the near-site attenuation parameter $\kappa_0$ = 0.028 s was adopted from [56]. We chose to select locations near epicentral area and set the epicentral distance to vary from 1 to 5 km. The corresponding target values of $PGA_{ROCK}$ were calculated in the range of 0.159–0.185 *g*. Based on these results, we estimated the median site amplification factor (±1 standard deviation) using the Stanko et al. [48] model for $V_{S30}$ categories defined by Eurocode 8 (Figure 9). Standard deviation of median site amplification describes site response variability as the effect of variations in ground motions and local site parameters. For site class B, we used two bins, 360–600 m/s (B1) and 600–800 m/s (B2), consistent with the findings of [57]. As underlined in [58], profiles with relatively shallow surface layers underlain by hard bedrock surfaces may have high impedance contrasts leading to strong resonance effects, which is the case for the Zagreb Podsljeme zone. The dominant period/frequency of the ground motion and the natural period/frequency of the soil are especially critical for possible resonance effects during earthquakes, which can result in heavy destruction. Therefore, knowing the building resonance frequency is important for structural stability during earthquakes and for limiting the extent of resonance damage. If the period of ground motion matches the natural resonance of a building, it undergoes the largest oscillations possible and suffers the greatest damage [30]. In general, the Zagreb residential area in terms of $V_{S30}$ categories is in categories B1 (Podsljeme) and C (city center and Novi Zagreb), which show site amplifications most prominent between 0.1–0.5 s (2–10 Hz) and 0.4–0.9 s (1.1–2.5 Hz). Compared to the period–building relationship of $T$ = 0.016$H$ presented in [59], these periods correspond to family houses and 3- to 4-story (B1) and 4- to 15-story buildings (C1).

The effects of the M5.5 Zagreb earthquake on low-rise and certain high-rise structures were similar, while differential effects are likely to occur in the case of higher-strength shaking (e.g., M7 or M8 events). The single-family dwelling type of short-period structures, i.e., buildings with $T < 0.2$ s [30], were mostly built in the post-1963 time period as confined masonry buildings. However, the damage that occurred was worse than it should have been. This could not be attributed to the PGA and the correspondence of their resonant period with the resonant period of the earthquake. It appeared that the severely damaged dwellings did not have proper post-1963 workmanship and construction (i.e., they were brittle and had irregular additions) or had poor foundations. However, this does not apply to the epicentral area, because there were severely damaged dwellings with the proper post-1963 or contemporary standard construction, i.e., designed for earthquake resistance. In determining the effects of earthquakes in the region, it is crucial to clearly define the structural system of the buildings so the effects can be properly differentiated. Seismic vulnerability assessment of single-family dwelling type of short-period structures should include several assessment steps: foundation systems, structure framing and configuration, condition of structure before and after an earthquake, nonstructural elements, age, size and, finally, the impact of site response that corresponds with observed damage (as was the case for the damage, classified as intensity 6, to one- or two-story family houses in the vicinity of the epicenter).

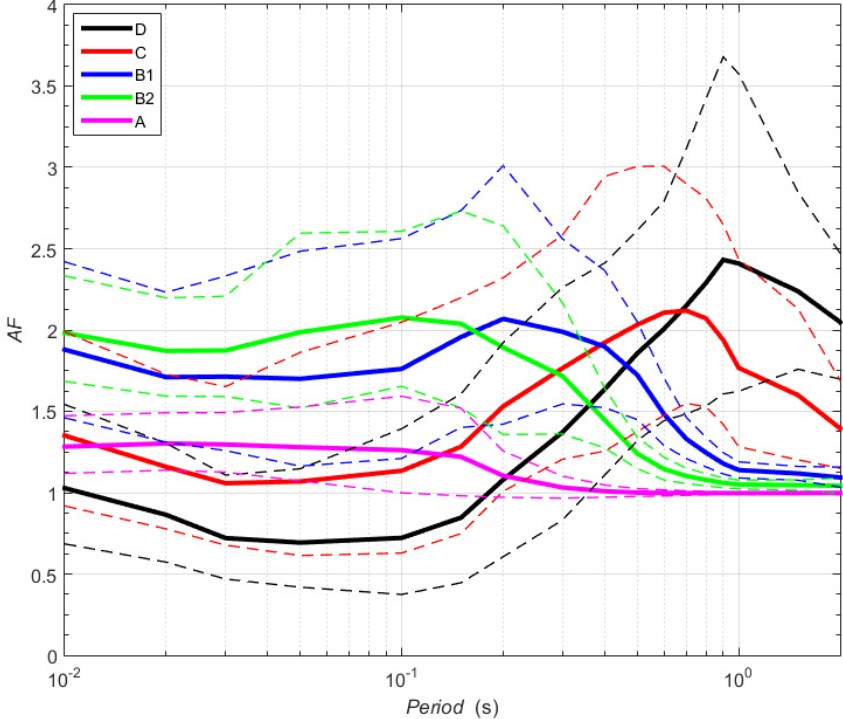

**Figure 9.** Median site amplification factors [41] for $V_{S30}$ site classes based on Eurocode 8 with (±1 standard deviation (dashed lines) for the $PGA_{ROCK}$ in the range of 0.159–0.185 *g*.

Figure 10 shows two site amplification maps for the *PGA* at the surface (AF@PGA) and for the predominant amplification peak period (AF@PP) for estimated $PGA_{ROCK}$. The amplification factor at the surface (AF@PGA) of the input M5.5 event seismic motion (approximately estimated between 0.16–0.19 *g*) tends to be 1.6–1.8 in the Podsljeme zone, approximately 1.4–1.6 in the city center and approximately 1.3 in the alluvial Sava zone (Novi Zagreb) due to the nonlinear effects of strong shaking [42]. At the resonant period (AF@PP), the observed amplification factor varies from 2.1 in the Podsljeme zone to 2.2 in city center and 2.4 in the Sava alluvial zone. Observed amplifications probably can also be correlated with ground motion polarization and directionality at sites with pronounced topography [60].

The first-order assessment of seismic amplification (due to site conditions) in the Zagreb area for the M5.5 earthquake shows that ground motions of approximately 0.16–0.19 *g* were amplified at least twice (if the standard deviation is taken into account, they can be amplified by a factor of 3.0 at some locations) in the Podsljeme and city center zones where the greatest extent of the damage is reported. Damage is most prominent in older buildings built before 1964, while newer buildings that were built using Eurocode 8 sustained very little or no damage, particularly in the Novi Zagreb area where the amplification factor at the resonant period is up to 2.4 (with standard deviation, it can be up to 3.6). Severe damage to the south tower of the Zagreb Cathedral (yellow square) could likely be the result of old construction and resonance effects; for category C, AF@PP = 2.2 at the period of 0.8–0.9 s (1.1–1.25 Hz), which corresponds closely to the Cathedral period of 0.71–0.92 s (1.08–1.4 Hz) [61]. Similar results are observed for the Croatian Parliament (also an old building that suffered considerable damage), where the soil category is B1 with estimated AF@PP = 2.08 at a period 0.31 s (3.16 Hz).

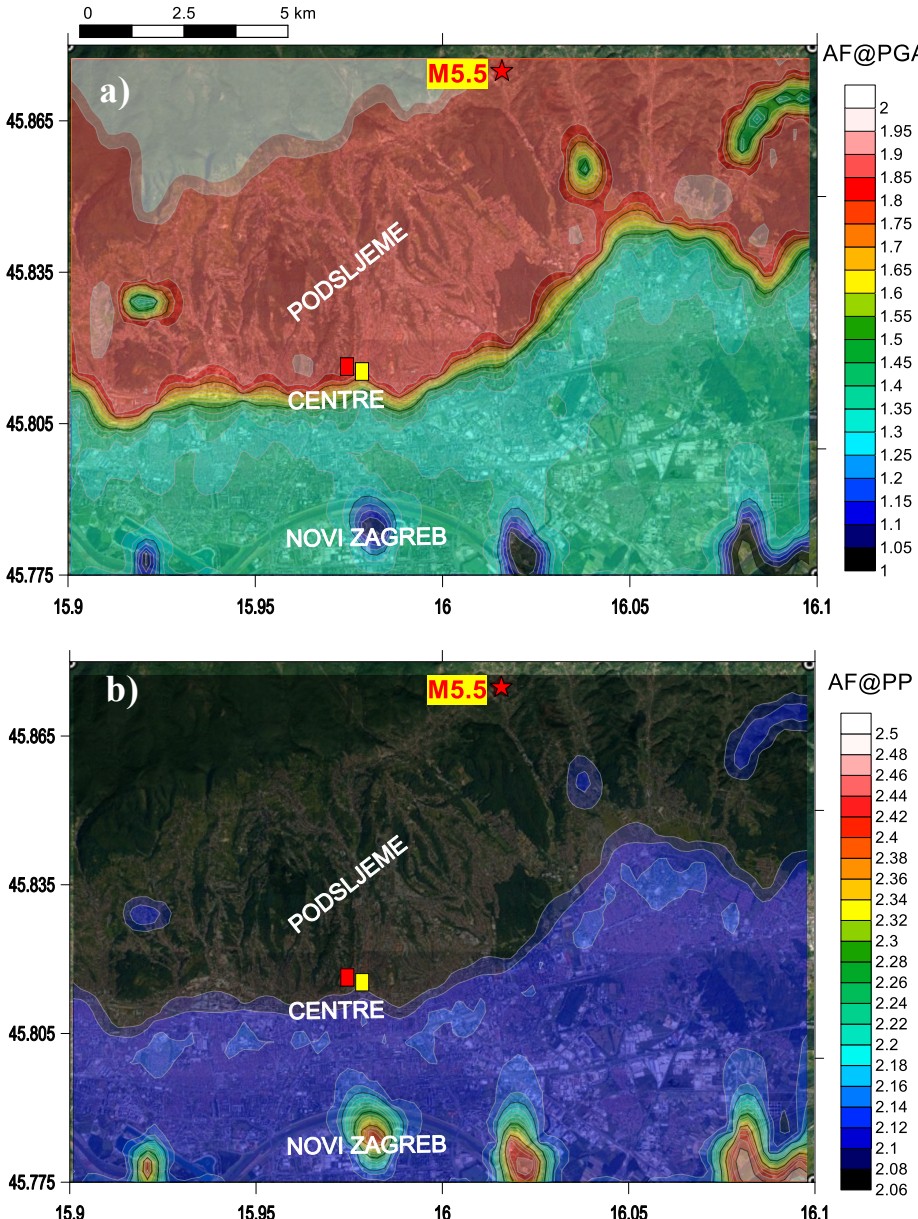

**Figure 10.** (**a**) Site amplification map for the *PGA* at surface (AF@PGA) and (**b**) the predominant amplification peak period (AF@PP) for estimated *PGA*ROCK.

As previously noted, buildings that were constructed after 1963 sustained little to no damage during the earthquake when compared to buildings constructed before 1920 (or even 1945). The enhancement associated with the increased ductility demand of the regulations in the post-1963 period, the construction materials and the time period of construction (before 1920, between 1920 and 1963, post-1963 and after Eurocode 8 regulations) with respect to building damage from available data certainly shows that the measures of earthquake-resistant design substantially reduce the damage to buildings, particularly in the locations with observed high amplifications. Knowing the site amplification at a particular spectral period should be applied to design earthquake-resistant structures to avoid potential resonance with seismic ground motion. Additionally, it is urgently necessary to reinforce existing structures, particularly old historical Zagreb buildings, to improve their seismic resistance and to take into account site amplification resonance effects (e.g., see [62,63]).

*4.4. Estimation of the Earthquake Co-seismic Vertical Ground Displacement Using Sentinel-1 Interferometric Wide-Swath Data*

The Sentinel-1 mission is the European Radar Observatory for the Copernicus joint initiative of the European Commission (EC) and the European Space Agency (ESA) comprising a constellation of two polar-orbiting satellites, operating in all weather conditions, at day and night, and performing C-band synthetic aperture radar imaging [64]. The mission is composed of two satellites, SENTINEL-1A and SENTINEL-1B, sharing the same orbital plane. The C-SAR instrument operates at a wavelength of 55.5 mm and supports operation in dual polarization. It includes a right-looking active phased array antenna providing fast scanning for elevation and azimuth. Sentinel-1 satellites operate in four exclusive acquisition modes: stripmap (SM), interferometric wide swath (IW), extra-wide swath (EW) and wave (WV) modes. The IW mode images consist of three sub-swaths using the terrain observation with progressive scans SAR (TOPSAR) technique [65]. With the TOPSAR steering technique, the beam in range is also electronically steered (backward to forward) in the azimuthal direction for each burst [66].

Two Sentinel-1A IW SLC images were processed to derive the co-seismic surface deformation at the Medvednica Mountains (Table 1). Space-borne SAR data are available on the Sentinel data hub [67]. The co-seismic displacement was investigated using differential interferometric synthetic aperture radar (DInSAR) technique [68]. The estimation of earthquake strength was based on the determination of the differential phase angle between two SAR images taken from almost the same satellite position at different times. The differential interferogram was filtered by relevant coherence (> 0.2). Interferogram coherence values were moderate on average (0.46), although the affected area is covered with thick vegetation. Time series processing and post-earthquake analysis using radar interferometry were performed with the open-source Sentinel Application Platform (SNAP). Deformation was referenced with the line-of-sight (LOS) direction from radar. The LOS data were acquired by merging different swath phases and then unwrapped using SNAPHU (Statistical-Cost Network-Flow Algorithm for Phase Unwrapping). The results of co-seismic deformation were geocoded to a WGS84 geographic coordinate system (Figure 11). Figure 11 shows an unwrapped interferogram from which it is possible to identify the most affected area.

**Table 1.** Overview of used satellite scenes.

| Sensor | Date | Track | Pass | Band |
|---|---|---|---|---|
| **Sentinel-1A** | 17 March 2020<br>23 March 2020 | 146 | Ascending | IW |

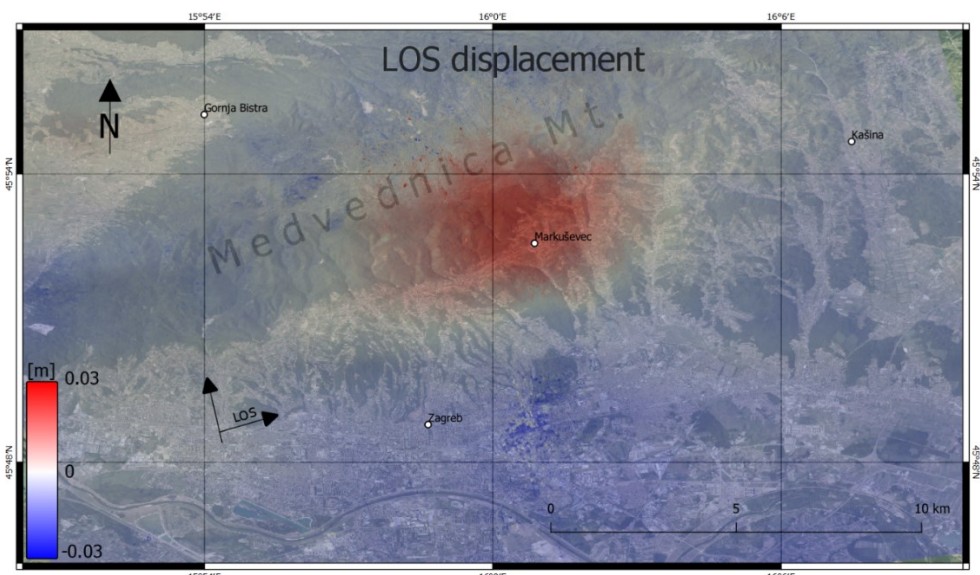

**Figure 11.** Unwrapped interferogram map indicating co-seismic line-of-sight (LOS) deformation of the 22 March 2020 earthquake.

The observed co-seismic deformation implies an approximately 3 cm uplift of the epicentral area that covers approximately 20 km$^2$.

*4.5. Preliminary Structural Modeling of the Zagreb 2020 Earthquake Sequence*

Seismological data collected during the main shock ($M_L$ = 5.5) and the following aftershocks, for the sequence from 22 March to 2 April 2020, provided input for preliminary structural modeling of the activated faults. The input data represented recorded hypocenters (points) comprising magnitudes, coordinates, depths and precise times of shocks. Based on the occurrence of the main shock and the aftershocks, time-lapse visualization of the first 24 hours of the Zagreb earthquake sequence 2020 was performed using the ESRI ArcScene software, version 10.2.1 (Supplement 1). The data showed that the main shock occurred below the city of Zagreb at a depth of 10.7 km and that the strongest aftershock occurred just a half-hour later and 2.7 km north from the main shock.

A set of 257 measurements was analyzed both spatially and temporally and processed for import into geological 3D modeling software. To achieve better visual insight, the hypocenters were scaled based on magnitude values ($M_L$) into four intervals: <2.0, 2.0–3.0, 3.0–4.0 and >4.0 (Figure 12).

Geological interpretation of the processed data included visual analysis, which allowed the extraction of points concentrated around possible fault planes, and selection of the datasets. The majority of the strong and moderate magnitude events occurred in first 24 hours after the main shock and were followed by hundreds of low and minor magnitude events (as shown in Supplement 1). Modeling of fault planes according to the datasets was performed with the MOVE 2019.1 software.

Based on the selected datasets, two major faults were modeled using inverse distance weight (IDW) interpolation of the preselected datasets (hypocenters). The faults represent the preliminary structural model of the Zagreb 2020 earthquake sequence (Figure 13).

Using the previously described methodology, a primary fault plane was modeled (Fault 1) based on the dataset containing 45 hypocenters, including most of the highest magnitude events in the Zagreb 2020 earthquake sequence. The fault is characterized by an orientation similar to the southeast-dipping instrumentally calculated fault-plane solutions of the main shock and the strongest aftershock (Figure 4). The rupture along Fault 1 propagated towards the northwest, apparently along a deep-rooted southeast-dipping thrust fault (125/31, dip direction/dip angle). The thrust fault can be projected to the surface where it virtually appears in the central part of the Hrvatsko Zagorje basin (Figures 1b and 13).

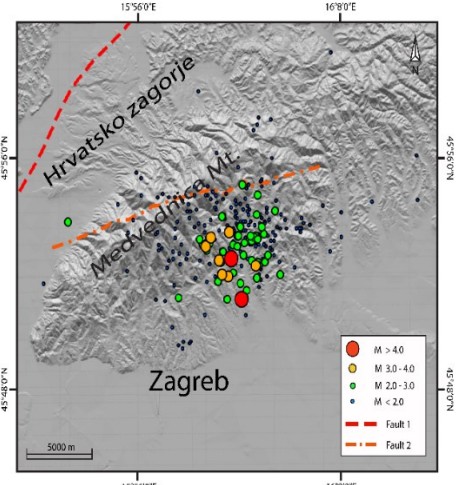

**Figure 12.** Spatial distribution of earthquake epicenter locations in the period from 22 March to 1 April 2020 (the Zagreb 2020 earthquake sequence). Fault lines are simple projections of the two modeled fault plains from the deep subsurface of the Medvednica Mountains to the surface (see Supplement 1 and Figure 13).

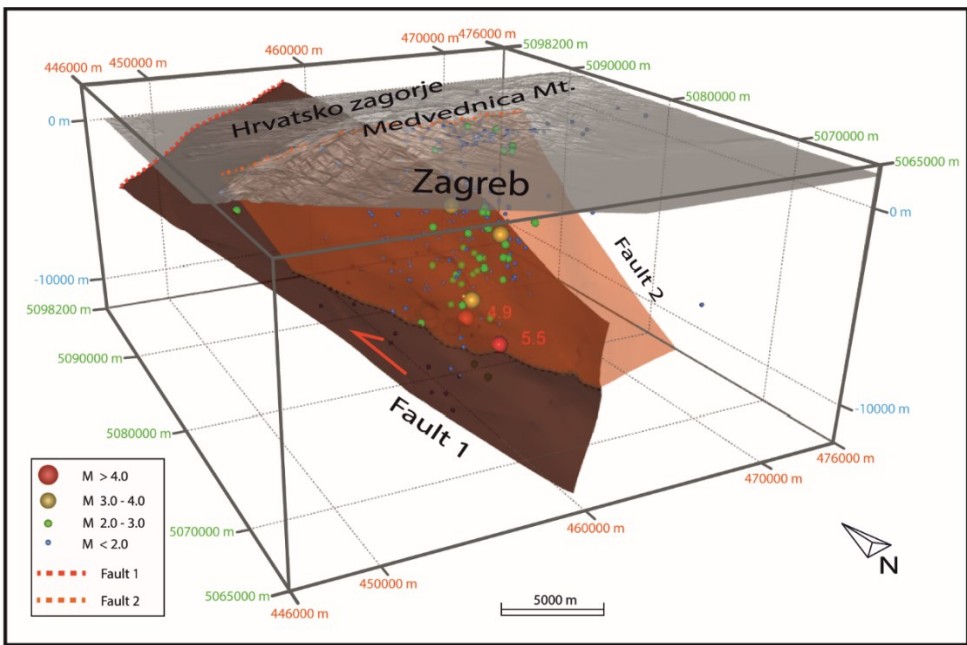

**Figure 13.** Preliminary structural 3D model of the Zagreb 2020 earthquake sequence. Coordinates are shown in HTRS96/TM coordinate system, and the elevation is in meters (no vertical exaggeration). Fault 1 is interpreted as a primary thrust fault, and Fault 2 is interpreted as a secondary (reverse?) fault.

In addition to Fault 1, a generally south-dipping fault (166/57) of unknown character (Fault 2) was modeled based on a dataset with 77 hypocenters situated above the main shock. Fault 2 includes most of the moderate magnitude events of the earthquake sequence.

A system of south-dipping longitudinal reverse faults has been mapped on the surface along the Medvednica Mountains [15] and interpreted in one of the latest tectonic models as the youngest (Pliocene to Quaternary) system of reverse faults ([11]; Figure 1) that are probably still active [7]. The reverse slip of the primary ruptured fault (Fault 1) of the Zagreb 2020 earthquake sequence (Figure 13; Supplement 1) generally corresponds to the frontal reverse fault striking along the northern foothills of the mountain (Figure 1b). However, the interpreted steep frontal reverse fault does not match the modeled gently inclined primary fault that is here interpreted as a thrust fault (Figure 13). Thus, a fault that could accommodate such a movement is probably a southeast-dipping thrust fault that completely dissects the basement of the central part of the Medvednica Mountains. This master thrust could represent a basal detachment (décollement) of the system of steeper reverse faults striking along the Medvednica Mountains. It could be steeper in the frontal part and thus could match the North Medvednica reverse fault, or it could be buried below the Neogene Hrvatsko Zagorje basin (Figure 1b; Figures 12 and 13).

The secondary fault (Fault 2) is a south-dipping, east–west striking fault of unknown character that dissects the mountain generally across the highest peaks (Figure 13). There are similar faults that are interpreted on the published geological maps as reverse faults [13,14], but further analyses and focused investigations are needed.

According to our results, the geological model of Van Gelder et al. [11] seems the most plausible for the main rupture of the 22 March 2020 Zagreb earthquake. The main shock occurred because of what we assume is a centennial-scale rupture of a segment of the thrust fault (Fault 1). The secondary fault (Fault 2) could represent either a steep reverse fault that belongs to the system or a tear fault that accommodates the movement of the tectonic block underlain by the primary ruptured thrust. According to a distinct surface co-seismic displacement of an area of approximately 20 km² (Figure 11), it could be assumed that the affected tectonic block in the hanging wall of both the thrust fault and the secondary fault moved up during the earthquake.

Other aftershocks occurred on unidentified faults that probably form a complex fault system. Further research is needed the interaction of the active fault system of the Medvednica Mountains to be reliably explained.

## 5. Conclusions

After the 22 March 2020 Zagreb earthquake, there were 26,197 damaged buildings in total (approximately 1900 uninhabitable), among which 9642 were family houses. The time period of construction with respect to damaged buildings revealed dramatic differences in their performances. The unreinforced masonry buildings from the time period before 1920 (or even 1945) had several times more damage than reinforced concrete (walls or infilled frames) or confined masonry buildings from the post-1963 time period. The damage was accompanied by widespread collapse of brittle chimneys.

The first-order assessment of seismic amplification (due to site conditions) in the Zagreb area for the M5.5 earthquake shows that ground motions of approximately 0.16–0.19 *g* were amplified at least twice (if standard deviation is taken into account, the amplification factor can reach 3.0 at some locations) in the Podsljeme and city center zones where the greatest extent of damage was reported.

Based on Sentinel-1 interferometric wide-swath data, the most affected area (with an uplift of about 3 cm) was identified, covering approximately 20 km$^2$.

Although the presented results are preliminary, it is evident that the M5.5 Zagreb earthquake in 2020 occurred on a reverse (thrust) fault that apparently dissects the basement of the Medvednica Mountains. The earthquake was most likely a consequence of tectonic activity, similar to the activity that caused the 1990 earthquake with an epicenter at Kraljev Vrh ($M_L$ = 5.0), since according to [7] the fault-plane solution of this earthquake almost coincides with that calculated for the 2020 earthquake.

The spatial and temporal analyses of the 22 March 2020 Zagreb earthquake indicate that the main shock and the first aftershocks occurred in the subsurface of the Medvednica Mountains along a deep-seated southeast-dipping thrust fault, recognized as a primary fault. The co-seismic rupture propagated during the first half-hour of the earthquake sequence along the thrust towards the northwest. Most of the aftershocks recorded during the first 24 hours occurred along a south-dipping, east–west striking secondary fault that is probably superimposed on the thrust fault. Other aftershocks occurred on unidentified faults that probably form a complex fault system.

This paper is a contribution to better defining the geodynamics of the complex fault system of the Medvednica Mountains and should be considered as a preliminary step towards better defining the input parameters for seismic hazard and risk assessment. In this regard, multidisciplinary research is necessary for the interactions of the fault system of the Medvednica Mountains to be reliably explained. In particular, focused geological mapping, more detailed spatial and temporal analyses of the complete seismic sequence and analysis of the co-seismic movements of the nodes involved in the geodynamic GPS network of the city of Zagreb will be object of future work.

**Supplementary Materials:** The following are available online at www.mdpi.com/2076-3263/10/7/252/s1: Supplement 1. The 22 March 2020 Zagreb earthquake sequence time lapse 3D visualization in ESRI ArcScene. Supplement 2. 3DPDF file of the initial structural model of the Zagreb 2020 earthquake sequence.

**Author Contributions:** S.M. conceived the presented idea, prepared the earthquake catalogue of the analyzed earthquake sequence, calculated fault-plane solutions, described the seismicity of the wider Zagreb area and prepared damage description. D.S. assessed seismic motion amplification. T.K. provided a geological background and interpretation of the results from a geological point of view, especially the conclusions based on time-lapse visualization. N.B. did structural modeling of the Zagreb 2020 earthquake sequence and prepared both supplements. D.P. described the performances of Zagreb buildings. B.K. estimated the co-seismic vertical ground displacement using satellite data. All authors discussed the results. S.M. took the lead in writing the paper, with great support from T.K. The other authors also contributed to the paper, provided critical feedback and helped shape the final paper.

**Funding:** This work has been supported in part by Croatian Science Foundation under the project HRZZ IP-2016-06-1854 and partly is the result of training and education conducted through GeoTwinn project that has

received funding from the European Union's Horizon 2020 research and innovation programme under grant agreement No. 809943.

**Acknowledgments:** The authors are thankful to the University of Zagreb, Faculty of Science, Department of Geophysics, Seismological Survey for providing the raw WM recordings of Croatian stations and especially to Tomislav Fiket and Ines Ivančić. We thank the editor, as well as the three anonymous reviewers for the constructive criticism on the submitted version of the manuscript.

**Conflicts of Interest:** The authors declare no conflict of interest. The funders had no role in the design of the study; in the collection, analyses, or interpretation of data; in the writing of the manuscript, or in the decision to publish the results.

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
