# Peer review of "The Zagreb (Croatia) M5.5 Earthquake on 22 March 2020"

_geosciences, doi:10.3390/geosciences10070252_

Round 1

Reviewer 1 Report

Overall a very heterogeneous piece of writing, with various scientific parts interspersed with more generic parts that seem more appropriate for a general earthquake report or a non-technical article. More tying together is needed between the contributions of the different authors in order to make an original research paper rather than an earthquake report or a collage of various elements. A better presentation and tying together of the parts will give a much more useful result.

An interesting narrative style is good. But be careful because sometimes the writing style is akin to a science popularisation article rather than a technical article. Similarly, some anecdotal information is nice but please keep in mind this is a scientific journal paper. Be careful with the shifts in style and ‘level’, please be more consistent.

The introduction does not seem to encompass the entire article (e.g. its motivation etc). It seems to stop a bit abruptly and deals with mostly tectonic issues. I think it should be rewritten in a more general ways covering all the range the article wishes to cover, and the specific tectonic details should go in the following section (which might be renames from geological to geological and tectonic overview?).

Fig. 1: where is Zagreb here? I found it in the end, but please make it more obvious, also make any key names bigger. Overall make it easier for the reader to connect the text with this figure. Note that the small print in the inset figure (bottom left) are not readable at all; please make that inset larger.

Section 3 needs more references (too much like a ‘story’ narration). Also, is there no macroseismic evidence for historical seismicity in the region, and no intensities to report? Please add more data to the general storyline.

Fig. 4 legend should not need an explanation of the beachball projections, the intended readership of “Geoscience” journal should be assumed to know such things. Again, an example of heterogeneity in compiling the article.

The CEC reference is Herak et al 1996 “updated” version – what is this? Is there no publication on Croatian catalogue in the 25 years since then? Also, is there no online resource? I think that “updated version from [15]” is not an acceptable reference.

Is there any information on historical seismicity from Ambraseys, Cecic or Suhadolc?

Line 170, why should waveforms be retrieved from IRIS and not the European data service structures?

Page 5: Are there any differences between the focal mechanism estimated by the authors and those published on major websites worldwide? Also, line 173: is there no additional evidence that can constrain the 2 alternatives? (in a later section, 4.5, there is more work on this – but the authors should foreshadow that in this early section, otherwise everything seems like pieces of a puzzle not properly put together). Also, are focal mechanisms really based on an ML=5.5 (i.e. local) with no Mw computed (knowing the saturation of ML at such high values)? or is the ML a typo?

Lines 182-183 read like an explanation to a non-scientist. Many parts of this paper give the same impression.

Fig 5 seems strange. We’d expect to see the seismicity plotted on a map rather than a pie-chart. Again, this style reminds one of a non-technical document. Overall, many things in the first part of this paper seem layman and addressed to non-scientists, and then in the last part there is a change of style that the reader has not been prepared for. There is a problem of consistency.

Where does macroseismic data come from for the 2020 events? Does Croatia have a ‘did you feel it’ system? What was sent to the EMSC service? Was there a formal reconnaissance squad in Zagreb studying building stock and failues? Wasn’t a map of damage distribution created by the Croatian team?

In Fig 6, are two of the photos taken form exactly the same place but different angles? If so then this is misleading and only one should be kept. Also it would be best to tag the photos to locations on a city map. More importantly, pictures like these (and many more) can be found here: https://www.emsc-csem.org/Earthquake/earthquake.php?id=840695#pics, so what makes this section more informative? The entire section 4.1 seems to depend (scientifically) more on the ‘foreign’ NASA data/map than on what the local scientists investigated. This is problematic. Statements such as “Citizens reported that in places around the epicenter, the 200 situation was very bad because of the destroyed houses” do not belong in a scientific journal. One expects more form this section, especially when authors come from engineering schools. (next section becomes very abruptly more technical and more structural-oriented)

Lines 235-242, describing design codes: Perhaps some design PGA values could be mentioned in addition to the change in rationale/materials/bearing member types. Also, Eurocode is only as strong as the national annex value onto which it is anchored (regional PGA), so again some relevant comments may be needed here.

Lines 230-231 need a reference.

Lines 243+: these comments are based on the study of all the 26000+ damaged buildings, or a subset? how were the conclusions drawn, what was the building stock sample?

It may be interesting and useful for some readers to make the connection between Mercalli and Gruenthal DG scales of intensity (are DG even necessary? are they used e.g. in Croatia? )

Lines 266-267: it is very important to suggest improvements the authors think are needed. Please state clearly where the possibility of a M6.5 in 475 years comes from (references etc). Also lines 263-280: very important, but where does it come from? some references are needed for the retrofit measures proposed, especially given some are indicated for special (historical/cultural value) buildings. Also references needed for the effects of Albania earthquake (ref 26 is from 2018, before the Albania event happened??). Overall this structural engineering section seems to have bee taken from another study, it is detailed but dos not present any evidence to demonstrate its validity.

Line 286: Kramer’s book is a bit too generic a reference for a journal article. However, Stur 1871 is a really interesting fact not well known, so it is worth mentioning – maybe even reproduce some of his sketches and findings, if available (if there was ever an opportunity to showcase his work it is probably in this article). But the paragraph that precedes the mention to Stur is a little too basic – the reader should know about source, path and site, linearity and nonlinearity from textbooks, no need to try and explain them here. Once more, heterogeneity in style and level.

In section 4.3 after line 310, a major leap is made. From previous sections where elementary notions were explained, random vibration and other issues are suddenly considered as obvious and the level increases very abruptly. It is evident here that different authors have written different parts and these parts were put together without great focus on coherence.

Line 335, what kind of buildings was this relation appropriate for? It has been stressed that Zagreb buildings come from different eras and different materials (also likely different plan views), so it is impossible to use a single empirical relation to get the resonant modes for all of the building stock??

Line 350: was there any topography mentioned in Zagreb? If mean Vs is as low as 200-500m/s as implied by the color map, then this seems unlikely?

How do the deformation levels found in section 4.4 tie in with the existing knowledge e.g. on tectonics etc? Also how does the displacement geographical distribution tie in with damage observations?

The first sentence is the conclusions (line 497-498) is actually not a conclusion, i.e. it was not mentioned or demonstrated in the article. Also, Line 524 says that this paper is about geodynamics – this is an example of the need to better integrate all the different pieces of the puzzle into a single coherent article.

The supplement material was not available.

Please specify all scales when mentioning magnitudes (e.g. section 3 on historic seismicity).

Please formulate structured paragraphs from the various ‘orphan' sentences in section 3 (and others).

Fig 2 needs an inset or additional plot to show where exactly we are on the map. Also where is Zagreb.

Fig 7 needs more explaining in the legend: what is each of the two figures what do red/orange colors mean etc? Also what does the big picture help with? isn’t all damage-related information given in the small one?

Fig 8 could use an additional scale inset in km. Also, if it is taken directly from USGS please make this even clearer in its legend.

reference for lines 37-39?

line 42: with M>? no sense in saying 661 earthquakes without a minimum value

line 244: then means than?

Author Response

Dear Reviewer,

thank you for your comments, corrections and suggestions to our manuscript. All comments have been considered carefully and the response to (or the action taken regarding) each comment is provided in the attached PDF file.

Best regards,
Snjezana Markusic

Author Response

Dear Reviewer,

thank you very much for your constructive review of our manuscript. All comments have been considered and the response to (or the action taken regarding) each comment is provided in the attached PDF file.

Best regards,
Snježana Markušić

Reviewer 3 Report

Dear Authors,

The manuscript is well written and clear. It provides good technical merit on complex Medvednica Mountains fault system. Although this is a small size event, it may help with our understanding the underlying crustal structure, site response and impact on the building performance better.

I suggest improvements on some figure resolutions such as Fig 4-5. Also, please add some scientific or statistical findings in the abstract to reveal the conclusions. Please provide Vs profile or generic profiles details in Section 4.3 while you explain Fig 9 (e.g. Fig 9a and 9b). It is interesting to see regional variations as compared to other areas if the audience wants to get a deeper understanding. Please help the audience understand how M5.5 type of event in the region can impact site response and single-family dwelling type of short period structures. In conclusions, please state the shortcomings of this study and/or future research.

Regards,

Author Response

(The authors gave the same response as above.)

Round 2

Reviewer 1 Report

The majority of recommendations made (with the exception of the structural engineering section) were not actually addressed by the authors.

Author Response

Dear Reviewer,

thank you for your comment. But I have to point out that all your comments were discussed, and accepted (constructive) suggested corrections were done. Also very
extensive response was prepared. From your 36 comments/corrections/questions (all listed in the answer sheet uploaded as our last response in PDF format): - 27 comments were fully or partially accepted, and all questions were answered, and -
on 9 comments were given argumented (in detail) explanation.

Best regards,
Snjezana Markusic